# High-Precision Trace Hydrogen Sensing by Multipass Raman Scattering

**DOI:** 10.3390/s23115171

**Published:** 2023-05-29

**Authors:** Jaspreet Singh, Andreas Muller

**Affiliations:** Physics Department, University of South Florida, Tampa, FL 33620, USA; jsingh4@usf.edu

**Keywords:** Raman scattering, trace detection, molecular hydrogen, optical cavities, multipass enhancement

## Abstract

Despite its growing importance in the energy generation and storage industry, the detection of hydrogen in trace concentrations remains challenging, as established optical absorption methods are ineffective in probing homonuclear diatomics. Besides indirect detection approaches using, e.g., chemically sensitized microdevices, Raman scattering has shown promise as an alternative direct method of unambiguous hydrogen chemical fingerprinting. We investigated the suitability of feedback-assisted multipass spontaneous Raman scattering for this task and examined the precision with which hydrogen can be sensed at concentrations below 2 parts per million. A limit of detection of 60, 30, and 20 parts per billion was obtained at a pressure of 0.2 MPa in a 10-min-long, 120-min-long, and 720-min-long measurement, respectively, with the lowest concentration probed being 75 parts per billion. Various methods of signal extraction were compared, including asymmetric multi-peak fitting, which allowed the resolution of concentration steps of 50 parts per billion, determining the ambient air hydrogen concentration with an uncertainty level of 20 parts per billion.

## 1. Introduction

Hydrogen (H${}_{2}$) is anticipated to be a major contributor to energy storage and power generation, potentially aiding decarbonization and sustainable development. Fuel cells have become increasingly sophisticated and can now power small devices such as computers, as well as larger systems such as vehicles. In principle, hydrogen can be produced from water electrolysis, preferably with surplus renewable electric power, though the majority of hydrogen production is still from fossil fuels. To help the hydrogen industry, trace sensors are necessary in monitoring inadvertent hydrogen losses. Therefore, portable and affordable H${}_{2}$ trace sensing devices, which can detect at or near the ambient hydrogen concentration of approximately 0.5 parts per million (ppm), are in high demand.

Despite its simple makeup, hydrogen gas remains difficult to detect at concentrations of a few ppm and below. Conventional optical methods such as cavity ring-down spectroscopy (CRDS), off-axis integrated cavity output spectroscopy (ICOS), and quantum cascade tunable infrared laser differential absorption spectroscopy (QC-TILDAS) are largely ineffective, as only quadrupole H${}_{2}$ transitions are infrared-active. Gas chromatography or magnetic resonance-based methods, while trace-sensitive, are bulky, expensive, and not readily portable. Thus, the most widely adopted H${}_{2}$ detection methods today use micro-electromechanical systems (MEMS) or metal oxide semiconductor (MOS) devices. These devices offer the advantages of high speeds, extreme miniaturization, and low costs [1]. However, poor chemical specificity and the need for recurrent calibration make accurate H${}_{2}$ detection at concentrations of order 100 parts per billion (ppb) a formidable challenge [2].

Recently, spontaneous Raman scattering (SRS) has become an attractive alternative due to its economical and robust nature. It is able to create a unique spectral signature for any molecular gas and, depending on the resolution, can detect dozens of analytes simultaneously, including isotopologues. Furthermore, enhancement techniques using capillaries, hollow-core photonic crystal waveguides, or optical cavities have been developed, significantly improving the prospects of SRS trace sensing applications. These techniques have demonstrated detection limits routinely in the 10 ppm range, and, in some cases, well below.

We report here the ultra-low-loss implementation of feedback-assisted multipass cavity-enhanced SRS applied to the trace detection of hydrogen. We examine various methods of extracting the H${}_{2}$ concentration at a level of 2 ppm and below. It is found that highly precise detection is possible, such that changes of 50 ppb can be systematically determined, even in complex gas mixtures such as ambient air or engine exhaust.

## 2. Background

As the simplest diatomic, hydrogen has played a unique historical role as a model for the validation of quantum mechanics molecular theory. In terms of its properties as a fuel, hydrogen has high energy content per unit of weight, making it an attractive option for transportation applications. The accurate measurement of hydrogen incorporation in solids is a matter of great interest in diverse research domains such as fuel cells, photocatalysts, and fusion reactors [3]. There has been a rapid increase in its application in the meteorological, aerospace, metallurgy, electronics, and chemical industries [4]. Outside its explosive range (4–75%), it may serve, for instance, in leak detection applications as a replacement for increasingly rare helium. At concentrations ranging from 4% to 75%, it is flammable, colorless, odorless, and thus potentially highly hazardous [5,6]. From the production, transportation, storage, and eventual utilization of hydrogen, it is essential to exercise caution and take appropriate measures to ensure the safe handling of this fuel [7,8]. Recent work has also pointed out the potential for hydrogen leaks to adversely affect the natural atmospheric removal mechanism of methane, further underlying the need for highly sensitive methods to quantify hydrogen concentrations in a variety of environments [9].

The primary requirements for a field-deployable H${}_{2}$ gas sensor are high sensitivity, a fast response, and stability over time. The traditional methods of gas chromatography (GC) coupled to mass spectrometry are difficult to implement for hydrogen. This is due to the fact that it is not possible to analyze a lighter gas than the carrier gas using gas chromatography coupled to mass spectrometry [10,11]. Hydrogen gas can only be analyzed by GC using a thermal conductivity detector (TCD), a helium ionization detector (HID), or an atomic emission detector (AED) [11,12]. Indirect detection of hydrogen gas by gas chromatography using electron ionization (EI) ion source mass spectrometry is possible [11,13,14]. Not only are these methods costly, but they are also time-consuming. Additionally, the equipment required for these techniques is bulky and demands skilled operators.

An alternative detection method is provided by chemically sensitized electronic transducer devices, relying on electrochemical or micro-mechanical readout [8]. Upon the deposition of hydrogen-absorbing and/or -adsorbing materials such as palladium and platinum, these sensors derive hydrogen concentrations via resistance for chemiresistors, or work functions for Schottky diodes, field-effect transistors (FETs), and metal oxide FET (MOSFET) sensors. Nanostructuring via, e.g., nanowires, can be further used to increase the surface-to-volume ratio to improve the sensing capabilities [15]. Additionally, resonator-based MEMS such as quartz crystal microbalance (QCM) and surface acoustic wave sensors have also been scrutinized extensively.

With a limit of detection (LOD) of 10 ppb, chemiresistors are one of the most effective hydrogen sensing devices. These sensors detect H${}_{2}$ gas by way of a change in the resistance of the semiconductor material [16] and feature a fast response (∼1 s) and recovery time (<10 s) [17]. However, a significant disadvantage is that their sensitivity is influenced by the temperature, typically optimal above 150 °C [16]. In contrast, Schottky diode sensors have a wide operating temperature range and exhibit excellent chemical stability. However, they suffer from a slow recovery time (>1 h) when exposed to certain gas atmospheres, including N${}_{2}$ [18]. Moreover, humidity adversely affects their performance [19]. MOSFET and FET sensors, although highly dependent on the gate morphology, are compact, highly stable, sensitive, and may have rapid response and recovery times [20]. MEMS-based surface acoustic wave sensors are efficient in mitigating the effects of temperature and pressure fluctuations but are highly affected by the surface properties of the sensing medium [21,22]. The underlying weakness of the entire family of electrochemical/electromechanical sensors is the intrinsically indirect nature of their detection mechanism, bringing about unavoidable nonspecificity and stringent calibration needs.

Spectroscopic optical techniques, by which a unique chemical fingerprint is probed, do not suffer from this drawback. However, optical infrared absorption modalities are not adequate for probing trace hydrogen, a diatomic homonuclear molecule that would only provide absorption via weak electrical quadrupole transitions [23]. Raman spectroscopy is an alternative optical technique that can be used for hydrogen detection. It is non-invasive with high selectivity, enabling hydrogen detection in complex gas mixtures [24,25,26,27]. However, due to scattering cross-sections in gases being only on the order of $10^{-31}$ cm${}^{2}$/str, additional experimental steps are critical to make Raman scattering viable for practical sensing. For example, stimulated photo-acoustic Raman spectroscopy (PARS), in which a high-power bichromatic laser source is combined with acoustic sensing, achieved hydrogen detection at a concentration of 10 ppm and an estimated LOD of 3.4 ppm at 0.3 MPa of pressure [28]. Aside from extreme sample pressurization, which has been implemented to detect hydrogen down to concentrations of 20 ppm at 2.5 MPa [29], numerous other enhancement approaches have been explored over the years. Among waveguide-based enhancement methods, fiber-enhanced Raman scattering (FERS) has shown particular promise. In hollow-core fibers as long as 2 m, hydrogen was probed with an estimated LOD of 4.7 ppm using either rotational or vibrational transitions [30,31]. Much progress in improving FERS has also been made recently by mitigating the issue of gas throughput by utilization of larger, anti-resonant, waveguides [32]. Another means of achieving Raman enhancement is using optical resonators. Besides Purcell-enhanced Raman scattering in microresonators [33], traditional low-loss resonant optical cavities may simply be used to create circulating power many orders of magnitude greater than the pump laser’s input power. Cavity-enhanced Raman scattering (CERS) has been implemented to detect hydrogen in high concentrations [26,34,35], and, recently, also at a 100 ppm concentration at 0.1 MPa of pressure, with an estimated LOD of 0.069 ppm, and at ambient levels (≈0.5 ppm), by way of circulating power as large as 1 kW [36]. Table 1 provides a summary of selected benchmarks reported for hydrogen sensing with different techniques.

Another powerful technique for SRS enhancement uses a non-resonant multipass cavity [37,38,39,40,41,42]. Unlike Raman enhancement methods involving a resonant cavity, multipass cavity Raman scattering requires no laser frequency stabilization or interferometric cavity length control. It further does not suffer from sample gas throughput limitations. Feedback-assisted multipass cavity enhancement with a laser diode has proven to be particularly effective (Figure 1) [27,43]. With an off-the-shelf multimode blue laser diode (NUBM44), several watts of power may be recirculated to obtain effective power at the sample of near 100 W. The feedback serves the additional purpose of reducing the laser linewidth to a few wavenumbers. With feedback-assisted multipass Raman scattering, hydrogen has been detected in ambient air and in breath samples, under pressurization of up to 0.8 MPa [40]. Nevertheless, a substantial limitation in earlier measurements was the background generated by the multipass mirrors themselves, precluding accurate detection in the ppb range. sensors-23-05171-t001_Table 1Table 1Summary of common techniques for detection of H2, with lowest detected concentration, LOD, and measurement time, when reported.MethodLowest Detected Conc. (ppm)Estimated LOD (ppm)Measurement Time (s)Reference**Chemiresistors
**500.01∼20[17]**Schottky diode
**10,0000.1∼10[44]**MEMS
**60.1∼2[45]**IR absorption
**5000 at 0.1 MPa10001[23]**Raman:
**



• PARS10 at 0.3 MPa3.4∼10[28]• FERS54.71–100[31]• CERS100 at 0.1 MPa
0.5 at 0.1 MPa0.069
-500
2500[36]• Multipass cavity0.075 at 0.2 MPa0.06600**This work
**


## 3. Instrumentation and Data Collection

The primary purpose of the present study is to examine the suitability of feedback-assisted spontaneous multipass Raman scattering for trace hydrogen concentration quantification. In particular, we aim to assess the types of capabilities that might be needed for a portable, economical, and durable device that could, for example, be permanently deployed outdoors in the vicinity of potential sources of hydrogen leakage. An enabling improvement to our setup was realized by employing ultra-low-loss mirrors to mitigate background noise contributions, thereby making high-precision hydrogen measurements possible. Custom ion-beam sputtered mirrors were acquired (Layertec 20030278), increasing the signal-to-background ratios (and therefore signal-to-noise ratios) by 24 times on average [27]. The remainder of our setup is similar to that in prior work [27]. Gas samples were introduced into a sealed chamber in which the multipass cavity resided, following a procedure that was identical for each sample. First, the chamber was evacuated to 0.07 MPa to remove as much background gas as possible (a lower pressure would potentially result in contamination due to outgassing). The chamber was then back-filled with the test gas to a pressure of 0.2 MPa. This pumping/purging process was repeated several times to guarantee high-purity sample loading. The loaded sample was dried by circulation through silica desiccant equipped with a particulate filter to a relative humidity of less than 0.1%. The drying, though not critical, aids in the removal of spectral interferences between the hydrogen Raman spectrum and the Raman spectrum of water vapor [43]. Care was exercised to ensure that the sample was loaded into the chamber free of contaminants, with a concentration stable over an extended period of time. Accordingly, the chamber seal was checked by observing the chamber pressure over time after pressurization to 0.2 MPa. With a total pressure loss of only 0.007 MPa over 12 h, the seal was such that contamination on a timescale of minutes remained undetectable. Indeed, upon loading the chamber with pure nitrogen, the concentration of stable air components such as methane or carbon dioxide was quantified by SRS, providing an estimate of sample purity of greater than 99.99%. Additionally, it was verified that that the desiccant did not measurably alter the concentration of gases relevant in the present study.

A diluted reference gas was obtained from Airgas (part X02AI99C15A48N1) with a certified H${}_{2}$ concentration of 1.961 ppm ± 5% in dry, hydrocarbon-free air. To obtain lower H${}_{2}$ concentrations, this reference gas was mixed with nitrogen (ultra-high-purity nitrogen from Airgas). Our focus was on the vibrational part of the H${}_{2}$ Raman spectrum, which contains multiple peaks within the range of 4100–4170 cm${}^{-1}$ (Figure 2). Although the differential scattering cross-section for pure rotational SRS is greater than that for vibrational SRS, the latter has the advantage of featuring peaks well separated from the peaks of other gases (except H${}_{2}$O), in the form of a rotationally resolved fingerprint (Figure 2a). The relative magnitude of the individual peaks being determined by a thermal distribution, the fingerprint will remain identical so long as the sample temperature stays constant. For this reason, the temperature at the center of the multipass cavity was maintained by a heater at 24 °C, with a variation of approximately 1 °C over 24 h. During a 10-min-long measurement at a pressure of 0.2 MPa, the main *J* = 1 peak could still be detected at concentrations below 100 parts per billion (Figure 2b). As we show below, this remarkable capability makes feedback-assisted multipass SRS uniquely able to quantify the hydrogen concentration precisely, even in complex gaseous environments.

## 4. Methodology for Quantification of Detected H${}_{2}$ Concentration

The most optimal means of extracting the H${}_{2}$ concentration from raw spectra such as those of Figure 2 can vary depending on the experimental conditions. We compared several methods in an attempt to minimize the variation in the outcomes of repeated measurements on a pure reference gas sample (H${}_{2}$ concentration of 1.961 ppm).

### 4.1. Numerical Curve Integral

The most straightforward method to estimate the peak area is by numerical summation over a predetermined number of detector bins (pixels). However, the accuracy of this method relies on prior determination of the peak’s baseline. The baseline may itself contain a spectral modulation. In principle, the baseline can be obtained in a separate measurement in which the analyte is absent. Besides doubling the measurement duration, however, this step could also introduce additional error. We, therefore, opted to determine the baseline by numerical interpolation of the background in the vicinity of the peak to second order in frequency.

### 4.2. Least-Squares Fitting (LSQF) Data Comparison

An effective method that forgoes the separate determination of the data baseline extracts the area in the data set of interest relative to the area in a reference data set associated with a sample of known concentration. In its simplest form, numerical sum-of-squares minimization yields the scale factor between the spectrum, $S_{\mathrm{TEST}}\left(\nu \right)$, from the sample of interest to the spectrum, $S_{\mathrm{REF}}\left(\nu \right)$, from the reference sample, i.e., the parameters *A* and *B* minimizing the quantity $\epsilon =\sum_{j}{\left|AS_{\mathrm{TEST}}\left(\nu_{j}\right)+B-S_{\mathrm{REF}}\left(\nu_{j}\right)\right|}^{2}$, where *j* runs over an interval such that $\nu_{j}\in$ ($\nu_{\min}$,$\nu_{\max}$), which includes only the peak of interest. The scale factor *A* represents the ratio of analyte concentration between the test and reference spectra, whereas *B* takes into account any baseline variation contribution. As an example, Figure 3 displays the result of a least-squares comparison between a sample with a nominal hydrogen concentration of 0.981 ppm and a commercial reference sample with a hydrogen concentration of 1.961 ppm. The extracted scale factor is 1.96, in excellent agreement with the nominally intended concentration. Although generally convenient, there are contexts in which this method is not preferred. For instance, if the test and reference spectra were not recorded under an identical excitation wavelength and spectral detection conditions, the frequency axis would also need a free adjustment parameter. Additionally, the residual noise in the reference spectrum will ultimately limit the precision with which the two spectra can be compared.

### 4.3. Gaussian Fitting Method

When the spectral lineshape under test is analytically known, least-squares fitting using the analytic function as a reference is preferred since it is, in principle, exact. For the spectrometer that we employ, the spectral response function closely matches a slightly asymmetric Gaussian. Therefore, the analytic function that we choose is a sum of four Gaussians, each of the form
(1)$$G\left(\nu ,\Gamma ,\nu_{0}\right)=\frac{1}{\Gamma }\sqrt{\frac{4\ln2}{\pi }}\exp\left[-4\ln2{\left(\frac{\nu -\nu_{0}}{\Gamma }\right)}^{2}\right]$$
where ν, $\nu_{0}$, and Γ denote the detection frequency, the line center frequency, and the full width at half maximum (FWHM), respectively. To introduce line asymmetry, we employ the function
(2)$$\Gamma =\Gamma \left(\nu ,\nu_{0},\Gamma_{0},a\right)=\frac{2\Gamma_{0}}{1+e^{a(\nu -\nu_{0})}}$$
where $\Gamma_{0}$ = 3.5 cm${}^{-1}$, and a=0.14 cm is a constant that quantifies the spectral asymmetry [46].

Figure 4 shows the asymmetric Gaussian fit obtained for the reference H${}_{2}$ (1.961 ppm) sample. To optimize the model parameters, a long-exposure (Figure 4b) spectrum is utilized. The error associated with the fitting parameter measuring the concentration in Figure 4a,b was 1.1% and 0.5%, respectively, providing the opportunity to detect variations in H${}_{2}$ concentrations of this order.

### 4.4. Normalization to Oxygen Concentration

We also consider the possibility of refining our estimate of the hydrogen concentration through normalization to oxygen or nitrogen, the most abundant species in the samples that we consider in the present work. This normalization can potentially correct for deviations in effective laser power or slight optical misalignments between different measurements. Particularly convenient as a reference is oxygen by way of its overtone peak at 3088 cm${}^{-1}$. To extract the oxygen concentration, however, method selection is necessary, and we compare a variety of combinations thereof below. As an example, Figure 5 shows a O${}_{2}$ overtone peak comparison between a sample and a reference spectrum using the least-squares data comparison method (Section 4.2).

### 4.5. Method Comparison

We conducted repeated measurements on the same sample (reference gas containing 1.961 ppm H${}_{2}$), which was freshly loaded before each measurement following the pump/purge procedure described above. The results of concentration extraction with different methods are shown in Figure 6. In part (a), no normalization was applied, i.e., no numerical factor corrected for any variations in experimental conditions, such as temperature variations in the room, that might slightly alter the optical beam paths in our setup. It can be seen that, on average, the curve integral and the Gaussian fitting methods feature less spread than the least-squares data comparison method. The spread is quantified in the form of a standard deviation and summarized for each method with and without different normalizations in Table 2. Surprisingly, the different methods yield outcomes that are highly uncorrelated. In fact, the average (light blue triangles) provides the lowest spread, revealing that the underlying error is random statistical rather than systematic. As expected, then, extraction methods that include normalization (Figure 6b,c) do not improve the precision, despite the high precision with which the normalization factor may be obtained, as evidenced by the data of Figure 6. On the contrary, the deviations seen in Figure 6b,c are greater than in Figure 6a because the normalization, with its own, though small, statistical random noise, amplifies the latter in the process of calculating a ratio. We conclude that for signal amplitudes of the type obtained in the range of 2 ppm H${}_{2}$, as in the data of Figure 6, the curve integral method is sufficient. However, with the H${}_{2}$ concentration reduced, as in the data below, it appears that the Gaussian fitting method provides the best result.

## 5. Results and Discussion

Using the asymmetric Gaussian fitting method without any normalization, we set out to determine the average precision by which a sample with an unknown H${}_{2}$ concentration can be characterized. Unless specified otherwise, measurements were performed at a total exposure (integration) time of 10 min, with the sample pressurized at 0.2 MPa (absolute).

We first examined the linearity of measurements for concentrations inferior to 2 ppm. Figure 7a shows a summary of several SRS spectra under dilution of the reference gas into nitrogen, resulting in the targeted H${}_{2}$ concentrations indicated. A measurement for pure nitrogen (green trace) is also shown. Upon extraction of the signal magnitude by Gaussian asymmetric fitting, we obtained the plot in Figure 7b for the extracted vs. the nominal concentration, found to be in excellent agreement. Although the present focus is on H${}_{2}$ concentrations inferior to 2 ppm, higher concentrations can and have been quantified by SRS in a linear manner also [31,47].

Crucially, minute changes in concentration can be differentiated. Figure 8 addresses this capability in a more detailed manner. In this case, the H${}_{2}$ concentration was varied in alternating steps from a base concentration of 1 ppm and 1.75 ppm. In both cases, the step was 50 ppb. As can be seen, this change is certainly detectable but also quantifiable so long as some repetition, i.e., averaging, is performed. The exposure time obviously is critical for the ultimate accuracy with which such concentration steps can be measured.

The exposure duration is also determinant in regard to the lowest concentration that may be detected and/or quantified, albeit at a diminishing return once the duration is on the order of several hours. This can be seen in the data of Figure 9, where the results of measurements at the same H${}_{2}$ concentration of 75 ppb are reported as a function of the exposure time. For shorter times, the spread between measurements is greater, as expected. However, there is little benefit in further increasing the exposure time beyond several hours as the precision gains are only marginal. Figure 9b,c show, in fact, that the noise in a 12-h-long exposure is only slightly lower than the noise in a 2-h-long exposure, due to square root scaling.

This random noise is ultimately what sets the LOD. With a definition of the LOD as the concentration level at which the peak signal exceeds five times the integrated noise level during the entire recording time, we estimated it at 60 ppb for 10 min, 30 ppb for 2 h, and 20 ppb for 12 h. Here, the Gaussian fitting proves to be much superior to the integral curve method to extract the concentration, so long as the ratio between the rotationally resolved peaks is maintained fixed, otherwise introducing considerable uncertainty.

As a last benchmark, we consider the task of determining precisely the concentration of hydrogen in ambient air. Increasing considerably in recent decades, the hydrogen concentration in air is often detected in atmospheric science studies by gas chromatography [48]. Our present study shows that spontaneous Raman scattering can be a competitive alternative for such a task. Figure 10 shows the outcome of alternating measurements of the H${}_{2}$ concentration of a sample of dry laboratory air and of the H${}_{2}$ concentration in the reference sample at 1.961 ppm. Consistent with the metrics reviewed above, the average H${}_{2}$ air concentration was determined to be 0.533 ± 0.02 ppm (1 standard deviation uncertainty), suggesting that in a more in-depth study, it should be possible to survey, for example, potential seasonal variations in the hydrogen concentration, if on the order of tens of ppb.

## 6. Conclusions

In conclusion, we have explored the potential of spontaneous Raman scattering for the purpose of precision molecular hydrogen detection and quantification. It is found that with ultra-low-loss mirrors, feedback-assisted multipass SRS is able to effectively detect step changes in H${}_{2}$ concentrations below 50 ppb in a 10-min-long exposure at 0.2 MPa of pressure. Quantification was optimally achieved at concentrations below 2 ppm by asymmetric Gaussian peak fitting of the entire rotationally resolved vibrational H${}_{2}$ Raman spectrum. The lowest concentration probed was 75 ppb, with a limit of detection of 50 ppb in a 10-min-long exposure. These results show that SRS holds promise as a portable characterization technique that can complement established laboratory analysis techniques such as gas chromatography. 

## Figures and Tables

**Figure 1 sensors-23-05171-f001:**
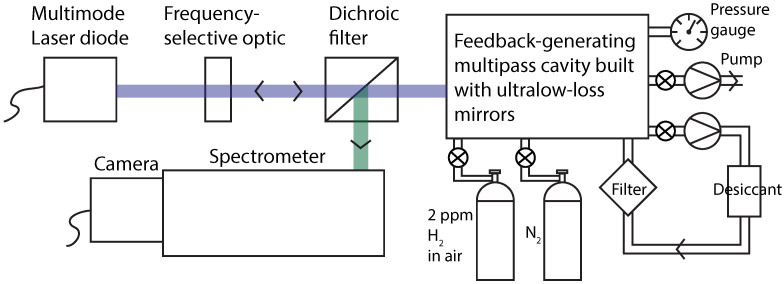
Diagram of experimental setup. The light from a multimode blue laser diode (Nichia NUBM44) is frequency-narrowed to $\Delta \bar{\nu }\approx$ 3 cm${}^{-1}$ by the feedback from an ultra-low-loss multipass Raman cavity using a frequency-selective optic (Optigrate volume Bragg grating). Feedback-assisted multipass SRS functions as an external cavity diode laser from which the Raman emission is dichroically coupled out to a spectrometer.

**Figure 2 sensors-23-05171-f002:**
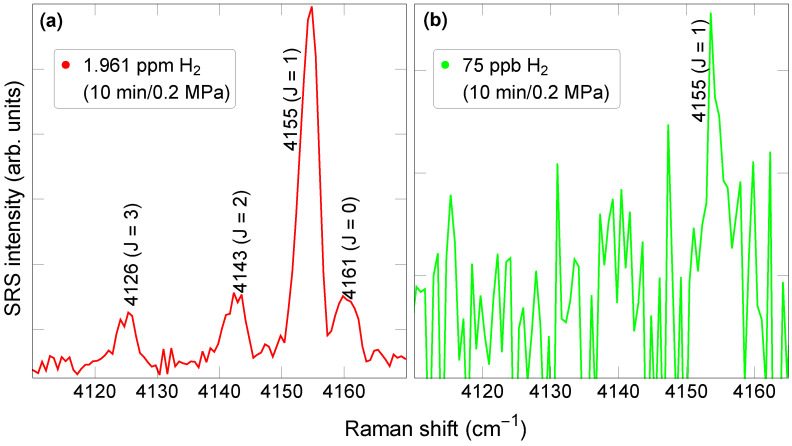
Raman spectrum with 10-min-long exposure for (**a**) 1.961 ppm nominal concentration (in red) and (**b**) 75 ppb nominal H${}_{2}$ concentration (in green). The pressure was 0.2 MPa.

**Figure 3 sensors-23-05171-f003:**
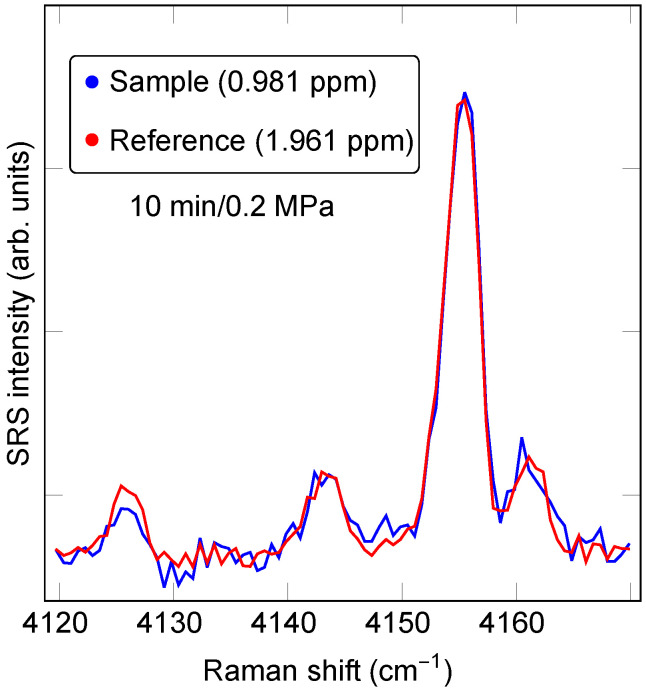
Least-squares data comparison fitting for relative H${}_{2}$ concentration determination at 0.2 MPa of pressure and 10-min-long exposure time. The least-squares-fit scale factor between the two data sets is 1.96.

**Figure 4 sensors-23-05171-f004:**
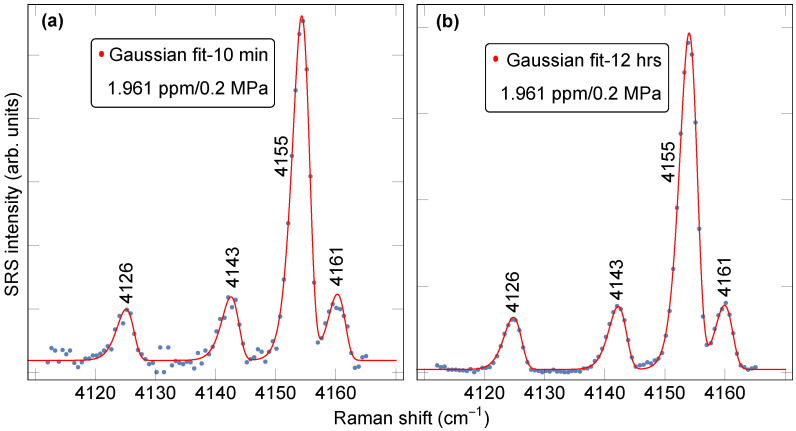
Asymmetric Gaussian fitting method for H${}_{2}$ concentration extraction, applied to the 1.961 ppm reference H${}_{2}$ sample spectrum recorded in a 10-min-long (**a**) and 12-h-long (**b**) exposure at 0.2 MPa pressure.

**Figure 5 sensors-23-05171-f005:**
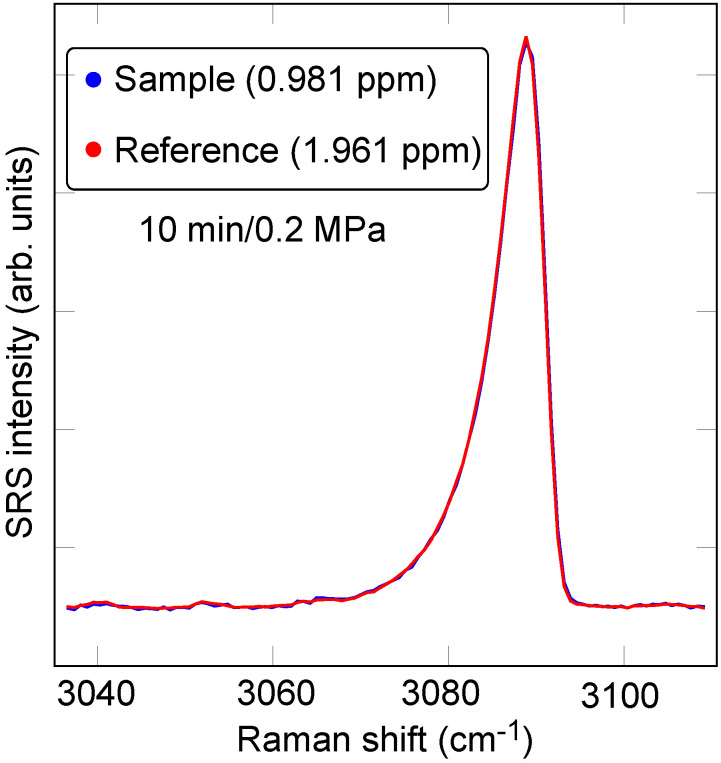
Least-squares data comparison of O${}_{2}$ overtone spectral feature as a potential means to correct for system fluctuations (laser intensity, alignment, etc.) by normalization of extracted H${}_{2}$ concentrations. The least-squares-fit scale factor between the two data sets is 2.003.

**Figure 6 sensors-23-05171-f006:**
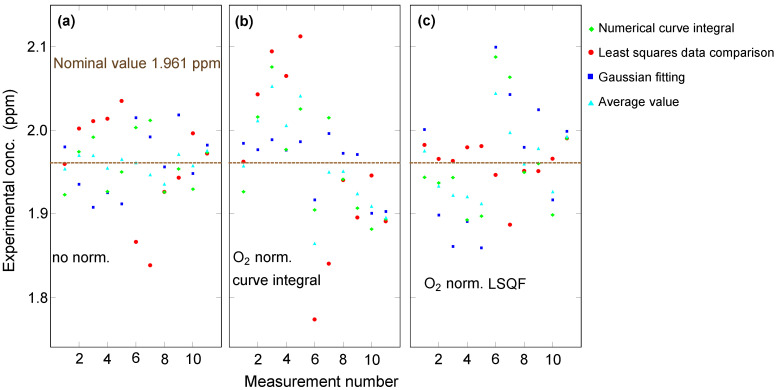
The concentration of H${}_{2}$ was extracted from vibrational SRS spectra using different methods. For the data in panel (**a**), no normalization was applied. For the data in panels (**b**,**c**), normalization by the O${}_{2}$ concentration extracted by the curve integral method and the least-squares comparison method was applied, respectively.

**Figure 7 sensors-23-05171-f007:**
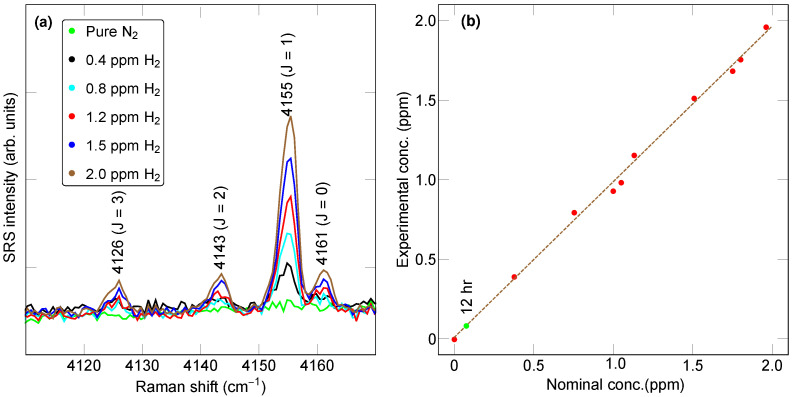
(**a**) Hydrogen Raman spectrum under various concentrations of H${}_{2}$ as indicated, obtained by diluting the reference gas in pure nitrogen. (**b**) Graph of experimentally obtained concentration as a function of nominally mixed H${}_{2}$ concentration. For the green data point, the exposure time was 12 h, while, for all the other points (red), it was 10 min. The pressure was 0.2 MPa.

**Figure 8 sensors-23-05171-f008:**
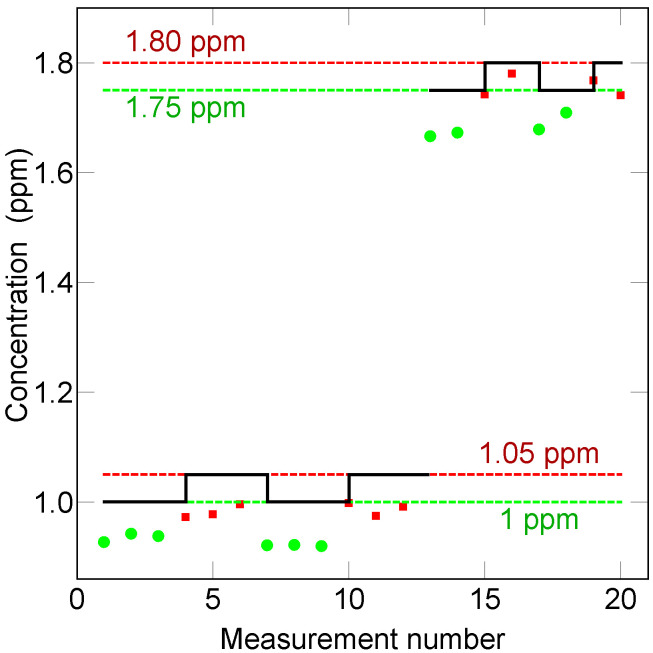
Measurement of hydrogen concentration steps of 50 ppb above a baseline of 1 ppm and 1.75 ppm. The exposure duration for each point was 10 min and the sample gas pressure was 0.2 MPa.

**Figure 9 sensors-23-05171-f009:**
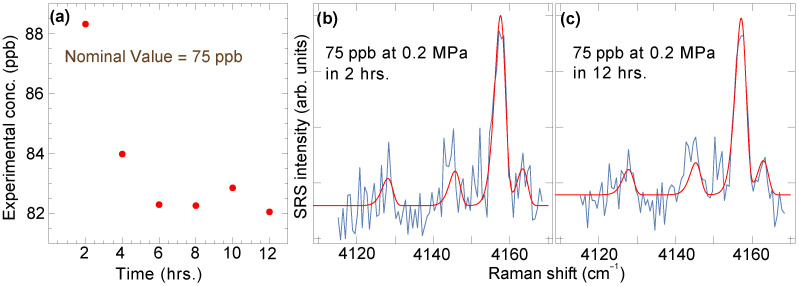
(**a**) Exposure-time-dependent measurements of H${}_{2}$ concentration in nominally 75 ppb H${}_{2}$ sample. Spectra and Gaussian fits for 2-h and 12-h exposures are shown in panels (**b**,**c**), respectively.

**Figure 10 sensors-23-05171-f010:**
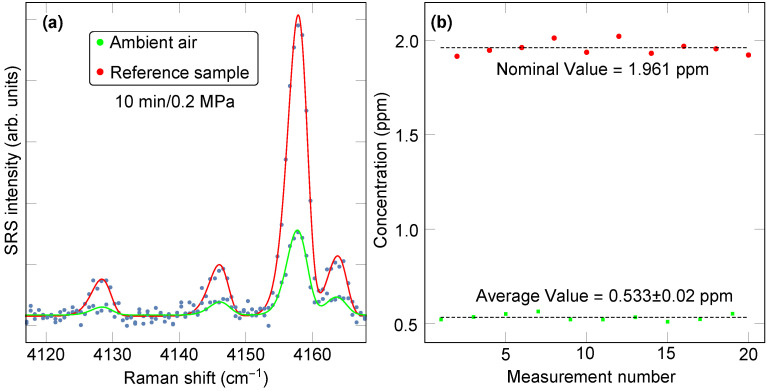
(**a**) Raman spectrum of ambient air (green) and 1.961 ppm reference sample (red) for 10-min-long exposure at 0.2 MPa of pressure. (**b**) Alternating measurements of ambient room air and the reference gas, which reveal the high precision and repeatability afforded by feedback-assisted multipass SRS.

**Table 2 sensors-23-05171-t002:** Summary of standard deviations, in ppm, extracted from the data shown in Figure 6, for each method of H${}_{2}$ concentration quantification.

Method	Numerical Curve Integral	LSQF Data Comparison	Gaussian Fitting	Average
Description	Summation over predefined number of bins around peak	Determination of scale factor between test and reference sample spectrum	Least-squares function fitting to retrieve peak amplitude	
No O${}_{2}$ normalization	0.03	0.06	0.04	0.01
O${}_{2}$ area normalization	0.06	0.1	0.03	0.06
O${}_{2}$ LSQF data comparison normalization	0.06	0.03	0.08	0.04

## Data Availability

The data presented in this study are available on request from the corresponding author.

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
