# Peer review of "High-Precision Trace Hydrogen Sensing by Multipass Raman Scattering"

_sensors, 2023, doi:10.3390/s23115171_

Round 1

Reviewer 1 Report

The article is well-written, provides solid background and good experimental results. It shows great potential of Raman spectroscopy to detect hydrogen, so I think it is suitable for publication in "Sensors".

I do not see any problems with data presentation or interpretation.

The only thing that requires attention is the reference numbering. Now they appear in wrong order (i.e. there is #18 in line 55, while the previous one was #2 in line 35), and some references seem to be missing (i.e. #9, #10).

Author Response

The only thing that requires attention is the reference numbering. Now they appear in wrong order (i.e. there is #18 in line 55, while the previous one was #2 in line 35), and some references seem to be missing (i.e. #9, #10).

We express our gratitude to the reviewer for bringing the matter to our attention. We have rectified the reference numbering within the text and eliminated the absent references.

Reviewer 2 Report

The manuscript reports the high precision sensing of trace hydrogen at concentrations below 2 parts-per-million through Raman scattering. The manuscript simultaneously provides a detailed description of various other analytical instruments that could be utilized to detect hydrogen and its plausible shortcomings. Overall, this is an interesting piece of work and needs to be published for broader attention. I, therefore, recommend publication of the manuscript with minor corrections.

1. What is the reason behind conducting the pumping/purging process with trace methane? Any other gas or so?

2. Is it possible to do the Raman quantification in the presence of an internal standard that can be used for scaling the signal intensities?

Author Response

  1. What is the reason behind conducting the pumping/purging process with trace methane? Any other gas or so?

We thank the reviewer for the positive response. The main objective of the pumping/purging procedure is to avoid the formation of a high vacuum within the chamber (thereby preventing the potential contamination caused by outgassing) while also removing most of the residual air in the chamber. In our original manuscript we may have introduced confusion by referring to methane codetection as a means of monitoring the purity of the sample gas loading, prior to the description of the samples probed. We simply meant to state that we have explicitly verified the purity of the sample loading process by Raman scattering measurements on other species such as methane which is present in air at the relatively fixed concentration of 1.9 ppm but is not present in our calibration reference sample. Alternatively, the chamber was simply loaded by high purity nitrogen and the presence of other gases such as carbon dioxide or oxygen present in air were monitored by Raman scattering. We have updated the text to have a more streamlined description of the sample loading process (see redlined manuscript).

  1. Is it possible to do the Raman quantification in the presence of an internal standard that can be used for scaling the signal intensities?

Yes, we believe it is possible to employ an internal standard for Raman quantification by scaling the signal intensities. As described in Fig. 6 and the accompanying discussion, we have conducted such scaling utilizing the O2 concentration but have arrived at the conclusion that unless substantial signal fluctuations occur (of order 10%), it is preferable to refrain from normalizing the intensity, as the scaling itself also introduces noise.

Reviewer 3 Report

The article describes a new spectroscopic method based on Raman scattering to detect ambient molecular hydrogen at leak concentration levels. Overall, the paper is well written and described the research well. Some experimental methods referred to a previous paper by the authors. It may be beneficial to provide a bit more details of the experimental schematic if possible

Author Response

It may be beneficial to provide a bit more details of the experimental schematic if possible.

We express our gratitude to the reviewer for the favorable feedback. We have incorporated an additional figure (Fig. 1) into the manuscript to provide a diagrammatic representation of the experimental setup for improved clarity.

Reviewer 4 Report

This manuscript entitled “High-precision trace hydrogen sensing by multipass Raman scattering” proposed a trace hydrogen sensor. A limit of detection of 60, 30, and 20 ppb was obtained at a pressure of 0.2 MPa in a 10-min-long, 120-min-long, and 720-min-long measurement. In addition, several methods of signal extraction were compared. I think this is a useful method worth to be published in Sensors. The manuscript is clearly written and the presented method is well supported by experimental measurements. I recommend it for publication after the following issues are addressed.

1.     In the paper, the detection limit of the hydrogen sensor is addressed clearly. However, the detection range is not described in detail. Please add the related parameters.

2.     The effect of environmental temperature on hydrogen detection should be researched more deeply. Please add the related parameters.

3.     Several methods of signal extraction were compared in the paper. A table should be added to summarize the methods.

Minor editing of English language language required. 

Author Response

  1. In the paper, the detection limit of the hydrogen sensor is addressed clearly. However, the detection range is not described in detail. Please add the related parameters.

We thank the reviewer for pointing this out. We have updated the text to indicate that the same instrument can also measure higher concentrations. We have added a citation to two sources that have explicitly described the linearity of SRS intensity with hydrogen concentration at much higher concentrations.  

  1. The effect of environmental temperature on hydrogen detection should be researched more deeply. Please add the related parameters.

                We are grateful to the reviewer for pointing out this important aspect. Indeed, the rovibrational fingerprint we probed is sensitive to the environmental temperature, which is why all our measurements have been performed at a constant sample temperature of 24°C enforced via a heater-based proportional/integral control loop. We have updated the text to include this information and the approximate stability (1C over 24 hours) of the temperature control system (see redlined manuscript for location of updates).

  1. Several methods of signal extraction were compared in the paper. A table should be added to summarize the methods.

                We agree with the reviewer that it would be beneficial to summarize the signal extraction methods. Therefore, we have now updated Table 2 with a dedicated row summarizing the methods of signal extraction.